# A Study on the Healing Performance of Mortar with Microcapsules Using Silicate-Based Inorganic Materials

**DOI:** 10.3390/ma15248907

**Published:** 2022-12-13

**Authors:** Cheol-Gyu Kim, Yun-Wang Choi, Sung Choi, Sung-Rok Oh

**Affiliations:** 1Department of Civil Engineering, Semyung University, Jecheon-si 27136, Chungcheongbuk-do, Republic of Korea; 2Department of Civil Engineering, KyungDong University, Yangju-si 11458, Gyeonggi-do, Republic of Korea; 3Research & Development Team, Newjust Co., Ltd., Gwangmyeong-si 14353, Gyeonggi-do, Republic of Korea

**Keywords:** healing performance, microcapsules, mortar, self-healing, silicate-based inorganic materials

## Abstract

Advancements in material science have led to the development of various self-healing concrete technologies. Among these is the use of microcapsule-based self-healing materials. This study evaluated the effects of self-healing microcapsules on the quality and healing properties of mortar. A silicate-based inorganic material mixture was used as the healing material tested with ordinary Portland cement. Accordingly, the effects of microcapsules (MCs) on the rheological, mechanical, and healing properties of mortar were determined. The mixing of MCs reduced the plastic viscosity and yield stress of the cement composite material owing to the particle properties of the MCs. The reduction was in proportion to the mixing ratio. The evaluation results show that the unit water permeability decreased owing to the healing reaction immediately after crack initiation. The healing rate was more than 95% at 7 days of healing age when more than 3% of MCs was mixed. This study provides a reference for the optimal mixing rate of MCs to achieve an ideal concrete healing rate.

## 1. Introduction

Recently, self-healing technology has been gaining popularity in reducing and repairing cracks in structures, mainly in developed countries. Self-healing technology can heal cracks on their own without any additional action in the early stages of cracks. Accordingly, it is possible to greatly reduce the cost, effort, and time required for general structure maintenance. In particular, since it is effective even when it is difficult for people to access [1,2], it is considered a very effective smart technology in the repair industry of structures. Crack self-healing technology has been initially studied in developed countries such as the UK, USA, and Japan [3,4], and recently, research results have been steadily reported in Korea, India, and China [5,6,7,8,9].

There are various methods for applying self-healing technology to structures. Self-healing technology is largely divided into autogenous healing technology and autonomous healing technology [10,11,12,13]. Autogenous healing is divided into natural healing and stimulated autogenous healing, and autonomous healing can be divided into bacteria, capsules, fibers, etc. [10,13,14,15]. Natural healing is literally the healing ability of the original structure [15,16]. Stimulated autogenous healing is a method of improving natural healing by adding organic/inorganic or hybrid materials [17,18,19,20,21].

Typically, crystal-growth-type materials such as mineral mixtures, expansion agents, and swelling agents are used [17,18,19,20,21]. Studies related to stimulated autogenous healing have been reported with the largest number of research cases among known self-healing techniques [22,23,24,25,26]. Looking at research cases related to stimulated autogenous healing, materials capable of crystal growth and reacting quickly are used [24,26]. Therefore, most of them are reported to have high crack healing performance, and studies are being conducted to heal larger cracks [27]. However, there is one downside. It has a technical limitation, namely, that the healing performance can be applied only to the early stage [10,11,12,13,27]. Most healing materials have a hydration reaction mechanism that reacts with water. Mineral materials used as healing materials are mixed in the process of manufacturing concrete. Concrete is an essential material that is mixed with water. In this case, the healing material starts to react slowly at the initial age when no cracks have occurred. Therefore, although the early age with reactivity has crack healing performance, in terms of long-term age, the reactivity is significantly lowered, so the healing rate is reduced or the healing performance cannot be expected compared to the healing performance of the early age [13,27]. Autonomous healing technology is a technology that improves these disadvantages. Representative techniques of autonomous healing include bacteria, capsules, and fibers. Fiber is a technology that uses engineered cementitious composites(ECC) to induce and heal large cracks into fine multi-cracks. Bacteria can also find many research cases. There are numerous results showing that bacteria have excellent healing properties [28,29,30,31,32,33]. However, from a long-term perspective, it is difficult to expect healing performance due to the limitation of food supply and demand for bacteria. However, the healing ability of bacteria has a longer shelf life than Autogenous healing technology. In the case of capsules, it is almost semi-permanent [13,27]. In the capsule, the core material, which is a healing material, is protected by the capsule membrane. Since the capsule reacts only when it is destroyed by cracking, it is free of time until cracking occurs. Accordingly, the healing energy of the capsule is semi-permanent. These features have the advantage of being able to respond to long-term cracks. A method using a capsule can compensate for these technical limitations and disadvantages [12]. This self-healing technology using capsules has two applications. The capsule can be mixed with the structure finishing material to coat the concrete surface, and it can be mixed with the concrete mixture to be used as a concrete matrix. In the former case, the cost is low because the amount of capsule used is relatively small; however, the healing performance is limited to the surface of the structure. In the latter case, the cost is relatively high due to the large amount of capsules used, but the healing performance is extended to the entire structure matrix. However, crack repair is more important than cost increase for structures with high importance or difficult to access by manpower. Therefore, the method of applying capsules for self-healing of cracks should be applied with comprehensive consideration of environmental and economic benefits.

If you look at the research cases related to self-healing capsules, there are relatively fewer cases than other types of research cases. The cornerstone of capsule technology was first tried at the University of Illinois in the United States in the early 2000s. Professor White’s research team at the University of Illinois studied self-healing cracks by encapsulating polymers [34]. A research team at Yonsei University in Korea researched microcapsules that can self-heal using sunlight (UV) without a catalyst [35]. Recently, self-healing technology using microcapsules or macrocapsules has been reported in many cases in China [10,27]. However, in conventional self-healing technology utilizing microcapsules, the core healing materials, such as epoxy, are mostly organic. Organic materials can produce effects in a short period. However, they cannot be integrated with inorganic materials, such as concrete, because of the difference in the modulus of elasticity. Consequently, it is difficult to ensure the long-term durability of crack-healing surfaces. In contrast, self-healing materials with inorganic cores can be integrated with concrete, improving the durability of the crack healing surface and maximizing the healing efficiency. In a similar case study, a research team at Pennsylvania State University microencapsulated water glass, an inorganic material, and mixed it with concrete to evaluate the healing performance. However, no additional research has been conducted since [36]. It is difficult to identify related research cases in Korea [37]. In addition, because microcapsules are mixed with the structure’s surface coating material and used for surface repair, the healing range is limited to the external surface microcrack damage [38,39]. Microcapsules should be mixed directly into the cement composite material instead of the surface coating material to extend the healing range. In this case, it is possible to expand the healing range because microcapsules capable of self-healing cracks are dispersed from the surface of the structure to the inside. Previous studies [40,41] showed that microcapsule mixing significantly reduced mechanical performance. However, further studies were needed to improve this [42,43,44,45]. Therefore, in this study, as a cornerstone study for direct mixing of microcapsules into concrete, a silicate-based inorganic material was microencapsulated, and the effect on the quality of mortar using it and the healing performance were examined.

## 2. Materials and Methods

### 2.1. Microcapsules

A silicate-based inorganic mixture was used as a self-healing microcapsule (MC). As shown in Table 1, the silicate-based inorganic mixture was mixed with three types of silicate at a certain ratio, and was used as a core material, a healing material. Table 1 shows the physical properties and chemical components of each silicate used as a healing material. For the composition ratio of silicate, the optimal composition ratio obtained through a previous study [37] was applied. Potassium silicate (K_2_SiO_3_): Sodium silicate (Na_2_SiO_3_): Lithium silicate (Li_2_SiO_3_) were mixed in a composition ratio of 5:4:1.

The MCs were encapsulated in the water/oil/water phase using an in situ polymerization method. The primary encapsulation was condensed in the water/oil phase by micro-dropping the healing material on general polyurethane (PU). PU has a strong viscosity and was dissolved by 60% using toluene. The second encapsulation was made by mixing the urea aqueous solution (U) and the first capsule to form an emulsion, and then encapsulated in the oil/water phase by adding formaldehyde (F). The encapsulation method was encapsulated through a generally known in situ polymerization method. Distilled water and a surfactant were used to improve the capsule yield, and resorcinol was used to strengthen the capsule membrane. In addition, sodium hydroxide and hydrochloric acid were used as pH regulators to create a capsule polymerization environment, and 1-octanol was used as an antifoaming agent to remove air bubbles. Finally, in order to reinforce the capsule membrane, a second capsule was added to an aqueous solution of tetraethyl orthosilicate (TEOS), followed by polymerization to coat the capsule membrane with silica. Monobasic sodium phosphate and tetrabutylammonium fluoride solutions were used as additives.

Figure 1a shows a sample of a silicate mixture, and Figure 1b shows a sample of an encapsulated silicate mixture. Figure 2 shows the encapsulation mechanism and Figure 3 shows the encapsulation process. Figure 4 shows the encapsulation manufacturing equipment. Equation (1) describes the healing mechanism of MCs. The silicate-based inorganic mixture, which is the core material of MCs, reacts with calcium hydroxide, which is abundant in the mortar, and alkali metal ions are gelled to produce calcium silicate hydrate and strong alkali ions. That is, cracks are closed by silicate-based reaction products.
Figure 1Microencapsulation of the core material as a healing material: (**a**) mixture of silicates; (**b**) encapsulated mixture of silicates.
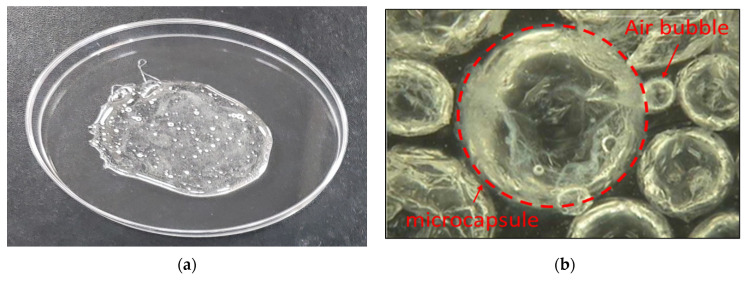

Figure 2Encapsulation mechanism of MCs.
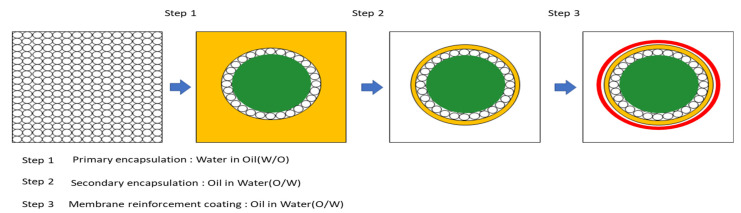

Figure 3Encapsulation process of MCs: (**a**) original method; (**b**) improved method.
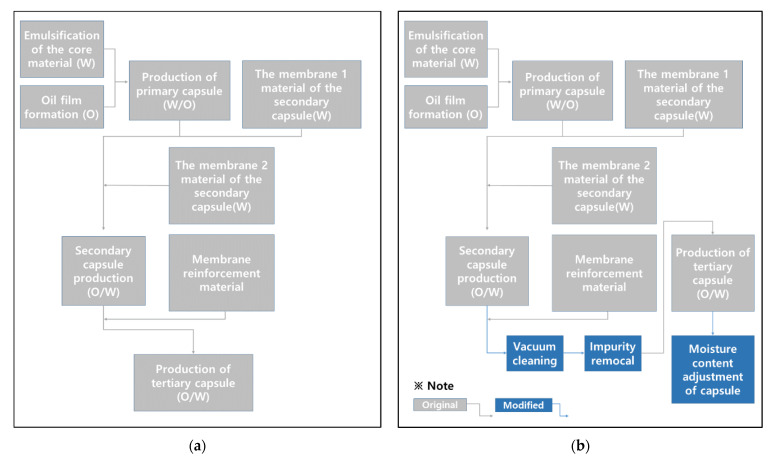

Figure 4Manufacturing device of MCs: (**a**) drawing; (**b**) automatic manufacturing equipment; (**c**) emulsion agitator and chemical mixer; (**d**) vacuum cleaners, automatic pH controllers, and particle size controllers.
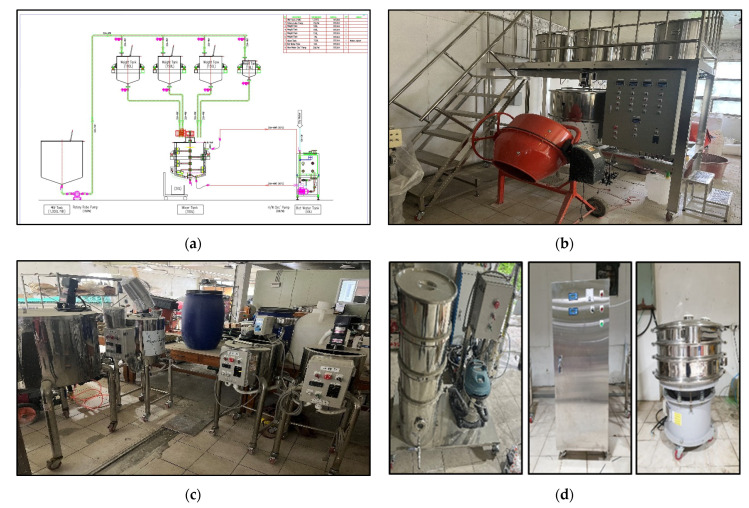

(1)$$A^{*}{}_{2}SiO+{xH}_{2}O\;+Ca{\left(OH\right)}_{2}\to \;CaSiO_{3}\;\cdot \;xH_{2}O+2AOH\;A^{*}=\mathrm{alkali}\;\mathrm{ions}\;(K^{+},\;{\mathrm{Na}}^{+},\;{\mathrm{Li}}^{+})$$
where,

$A_{2}SiO_{2}+{xH}_{2}O\;$: silicate minerals as core material in microcapsules

$Ca{\left(OH\right)}_{2}$: hydrates of cementitious materials

$CaSiO_{3}\;\cdot \;xH_{2}O$: calcium silicate hydrate(C-S-H)

2AOH: alkali ions

The size of MCs is determined by the stirring speed (RPM) during polymerization. For example, on average, MCs of about 350 μm at 50 RPM, about 270 μm at 150 RPM, about 210 μm at 250 RPM, and about 180 μm at 350 RPM are produced. In this paper, MCs were polymerized by applying the optimal stirring speed obtained through previous studies. The stirring speed was 50 RPM, and MCs with a size of 300 to 350 μm were used. Figure 5 shows a sample of MCs.

### 2.2. Mortar

Table 2 shows the mixing ratio of cement mortar. Ordinary Portland cement (C) was used for cement, and ISO standard sand (S) was used for fine aggregate. Table 3 will show the particle size distribution of ISO standard sand. In addition, in order to secure the target fluidity of the cement mortar, a polycarboxylic high-performance water reducing agent (ad.) was used. ‘ad.’ can reduce the loss of MCs by improving the workability of the mortar, and facilitates the dispersion of the MCs. MCs were mixed at 1, 3, and 5 mass% of cement mass.

### 2.3. Quality Properties

#### 2.3.1. Fresh State Test

For rheology, a mortar viscometer with a modified chamber size of brookfield’s paste viscometer DV-III ultra-model was used to measure the plastic viscosity and yield stress.

The shear rate was set to a descending step method and measured to avoid hysteresis of the measured value owing to the loop area. The measured shear stress was analyzed using the Bingham model through linear regression analysis. Figure 6 shows the rheological analysis method. Figure 6a shows the mortar viscometer, Figure 6b shows the hysteresis curve and Figure 6c shows the Bingham model for rheological analysis.

The flow of the mortar was measured using a flow table and flow cone according to the American society for testing and materials (ASTM) C1437/C1437M [46]. Mortar flow was measured in three directions, and then the average value was obtained. The air content of the mortar was measured according to ASTM C231/C231M [47].

#### 2.3.2. Hardening State Test

The compressive and flexural strengths of the mortar were measured according to ASTM C109/C109M [48] and ASTM C78/C78M [49], respectively. According to ISO 679, a test piece of 40 mm × 40 mm × 160 mm is prepared, and the flexural strength is measured first. After the flexural strength is measured, the remaining two pieces are used as test pieces for measuring the compressive strength. Therefore, as for the number of test pieces, three flexural strength test pieces are molded, and compressive strength measurement is performed using 6 fractured pieces of the flexural strength test piece. A general hydraulic universal testing machine (UTM) (Heungjin test machine, Gyeonggi-do, Republic of Korea) was used as the testing equipment for strength testing. Figure 7a shows the flexural strength test view, and Figure 7b shows the compressive strength test view.

### 2.4. Self-Healing Properties

#### 2.4.1. Water Flow Test

In the water flow test, the water flow rate (mL/min·mm) of the crack-induced specimen was measured to evaluate the self-healing performance. A 50 mm × 5 mm specimen was prepared and cured in water at a temperature of (20 ± 3) °C and humidity of (50 ± 10)% for 28 days. Subsequently, artificial penetration cracking was induced. Cracking was controlled by inserting a crack-inducing film such that the crack width was 0.15–0.3 mm. Thereafter, the side of the test piece was fixed with an iron frame to maintain the crack width and prevent fluctuations. The crack-induced specimens were cured in a humid chamber at a temperature of (20 ± 3) °C and humidity of (70 ± 10)%. The curing period was defined as the healing period. The healing test measured the initial water flow rate one day after the healing period [50,51,52]. This is because the core material of the MCs is in a liquid state; thus, a reaction time is necessary. Therefore, the water flow rate was measured after 1 day. The measured value was used as the initial reference value. The water flow rate for each healing period was compared at 3 days, 7 days, and 14 days.

For the test conditions, the water flow rate for the first 5 min was excluded to ensure that the water state of the specimen was the same. The water flow rate for the next 10 min was used for the analysis. The water flow rate was measured by connecting an electronic scale to a computer to measure the real-time water flow rate. Figure 8 shows the water flow specimen. Figure 9 shows the crack induction and specimen control process. Figure 10 shows the test equipment.

The healing rate ($H_{R}$) is defined as the rate of decrease in the water flow rate ($MCs_{0}\left(t\right)$) of MCs in the healing period (t) with respect to the water flow rate ($P_{0}$) of MCs 0% (Control). For the healing rate, the reduction level of the water flow rate on day (n) was obtained from the water flow rate on day 1 [53].
(2)$$H_{R}=\left[1-\frac{MCs_{0}\left(t\right)}{P_{0}}\right]\times 100,$$
where $H_{R}$ is the healing rate (%)

$P_{0}$ is the water flow rate of MCs 0% (Control).

$MCs_{0}\left(t\right)$ is the water flow rate of MCs during the healing period (t).

#### 2.4.2. Crack Closing Test

Crack closing was observed at a magnification of ×100 using the same test piece as the water-flow specimen. Healing products in cracks were also observed. Figure 11 shows the front view of the crack closing test.

## 3. Results and Discussion

### 3.1. Properties of Fresh Cement Composites

#### 3.1.1. Rheology

Table 4 shows the rheological parameters of the mortar. As MCs increased, the plastic viscosity, the slope of the Bingham model, decreased, and the yield stress, the y-intercept, also decreased. The plastic viscosity of Plain was approximately 24.5 Pa·s, and the reduction rates for MCs 1%, MCs 3% and MCs 5% were 9.4%, 15.9% and 26.1%. In addition, the yield stress of the plain was approximately 211 Pa, and the reduction rates of MCs 1%, MCs 3% and MCs 5% were 28.0%, 21.8%, and 17.5%. The yield stress of the MCs 1% decreased by approximately 25% but increased slightly as the mixing ratio increased. According to these results, it is inferred that the viscosity decreased according to the ball bearing effect because the MCs grain is spherical. Through these results, it is expected that the flow of the mortar will be reduced by the plastic viscosity, and the workability will be improved by the yield stress related to the thixotropy of the mortar.

#### 3.1.2. Flowability and Air Content

Table 5 shows the test results of the mortar flow and air content. Plain’s flow was approximately 180 mm and tended to decrease with increasing MCs. The flows of MCs 1%, MCs 3% and MCs 5% decreased by 5.6%, 11.1% and 16.7% compared with the flow of Plain, approximately. These result, according to a previous study [37,40], the cause of the decrease in the flow was reported as the loss of MCs of approximately 10%. In this case, the flow was reduced as the MCs were destroyed and acted as an alkaline activator in the mix process. These results indicate that the effect of the loss of MCs on quality is more dominant than the effect of the particle characteristics of MCs. In addition, it is believed that there is also a decrease in flow because the plastic viscosity decreases.

The air content of MCs 0% was approximately 6%. MCs 1%, MCs 3%, and MCs 5% were found to have the same air content regardless of the mixing ratio. These results indicate that the effect of MCs on the amount of air in mortar was insignificant based on the air amount error range of ±1.5%.

### 3.2. Mechanical Properties

#### 3.2.1. Compressive Strength

Figure 12 shows the compressive strength of mortar according to age. The compressive strength of Plain at 28 days of age was 47 MPa, and the compressive strength of MCs 1%, MCs 3% and MCs 5% decreased by 1%, 2% and 3% compared to the compressive strength of Plain, approximately. These results are considered to be that the compressive strength decreases because MCs are weak, and as the MCs increase, the compressive strength decreases proportionally. However, since the reduction level of compressive strength is not large within the error range, it is considered not to be significant, and it satisfied more than the target strength.

#### 3.2.2. Flexural Strength

Figure 13 shows the result of measuring the flexural strength of the mortar according to the age. The flexural strength showed the same tendency as the compressive strength evaluation result, and the level of strength development according to age was almost similar. The flexural strength of Plain at 28 days of age was approximately 9.6 MPa, and the flexural strength of MCs 1%, MCs 3% and MCs 5% decreased by 3.1%, 4.2% and 5.2% compared to Plain, approximately. However, because the reduction was less than 1 MPa, the MCs mixing ratio effect on the flexural strength was considered insignificant.

### 3.3. Self-Healing Properties

#### 3.3.1. Water Flow

Table 6 shows the water permeability and healing rate of mortar according to the water flow test. The crack width of the crack induction test specimen was induced to 0.2 mm at the age of 28 days. The crack induction specimen was subjected to a healing period of 1 day and then the initial water flow rate was measured. All types showed that the water flow rate decreased and healed as the healing period elapsed. These results suggest that even in the case of Plain (MCs 0%), there is a healing performance. In other words, it suggests that a certain portion of cracks can be healed by the natural healing effect even in general structures, and that the natural healing performance can be improved through MCs. It can be analyzed that the natural healing effect is due to the unhydrated cement present in the crack. These results mean that the initial crack width can be reduced by the healing effect and the crack width can be prevented from expanding. In this study, the re-sults were obtained for through cracks. However, cracks in real environments gradually expand from microscopic cracks. Therefore, healing the initial microcracks means that it is possible to reduce large cracks after all, and the healing effect increases as the microcracks grow. In addition, MCs used to heal microcracks react only around microcracks, which means that it is possible to respond to cracks that occur later. As for the effect of MCs, the healing rate tended to increase as the number of MCs increased. The initial water flow rate of MCs 1%, MCs 3% and MCs 5% decreased by 74.2%, 84.8% and 91.7% compared to plain, approximately. After 14 days of healing, MCs 1% was 86.1%. In addition, MCs 3% was 84.8%, and MCs 5% was 91.7% at 3 days of healing period. MCs 3% and MCs 5% showed a 100% healing rate with almost cracks healed within 7 days of healing.

Figure 14 shows the correlation between the healing period and water flow using the data in Table 6. Figure 15 shows the time required for healing for each type. Figure 16 shows the correlation between healing period and healing rate. The water flow of MCs 3% and 5% was reduced by approximately 97% or more compared with the water flow of MCs 1%. The results showed that the water flow decreased as the healing period increased when the MCs were mixed. Moreover, the results showed a proportional decrease as the MCs mixing rate increased. These results indicate that the healing efficiency increased as the MCs mixing rate increased. However, the optimal mixing ratio should be considered from another point of view. When a large number of capsules are mixed to heal a large crack width, the initial cost increases. Therefore, it may be inefficient compared with the healing efficiency. Natural cracks are generally formed by microcracks and then gradually expanded. Therefore, even if the MC mixing rate is low, the healing performance and healing rate will be higher if the healing action is performed immediately after the occurrence. 

#### 3.3.2. Crack Closing

Figure 17, Figure 18, Figure 19 and Figure 20 show the surface crack monitoring results of crack-induced specimens. In the case of MCs 1%, partial healing products could be observed, and in the case of MCs 3% and MCs 5%, the surface cracks were completely closed. Figure 21 shows the results of observation of healing products on the crack surface. Partially undestroyed MCs could be observed, and destroyed MCs and capsular membranes were identified. Furthermore, it was confirmed that the cracked area was repaired with a silicate-based product.

## 4. Conclusions

In this study, as a cornerstone study for direct mixing of microcapsules into concrete, a silicate-based inorganic material was microencapsulated, and the effects on the quality of mortar using it and the healing performance were examined. The following conclusions could be drawn.

According to the results of the analysis on the effects of microcapsules on the rheological properties of mortar, the plastic viscosity, and yield stress were reduced because of the particle properties of the microcapsule. Furthermore, the results showed a decrease in proportion to the microcapsule mixing rate. However, because the core material of the broken microcapsules acted as an alkali activator, the table flow decreased. These results are believed to be more influenced by the loss than by the particle characteristics.As a result of evaluating the effect of microcapsules on the mechanical properties of mortar, the strength decreased as the microcapsules were mixed, and the result decreased proportionally as the mixing rate increased. However, the level of reduction is not large, so it is considered that the target performance can be secured.As a result of evaluating the effect of microcapsules on the healing properties of mortar, it was confirmed that Plain without microcapsules had natural healing performance, but the healing rate was up to 75.9%, approximately. It is thought that the effect of microcapsules can promote natural healing performance and improve healing rate, and it was found that the healing rate and healing rate accelerated as the mixing rate increased.

Based on these results, the optimal mixing rate of microcapsules was determined. This rate can be used to mix large amounts of microcapsules with cement composites to heal large cracks. However, this is impractical because it would increase the cost. Therefore, it is necessary to control microcracks at the early stage of cracking to prevent the propagation and induce a healing reaction. When considering natural cracks, the healing efficiency is believed to increase even if fewer microcapsules are used; therefore, additional review on this matter is necessary.

## Figures and Tables

**Figure 5 materials-15-08907-f005:**
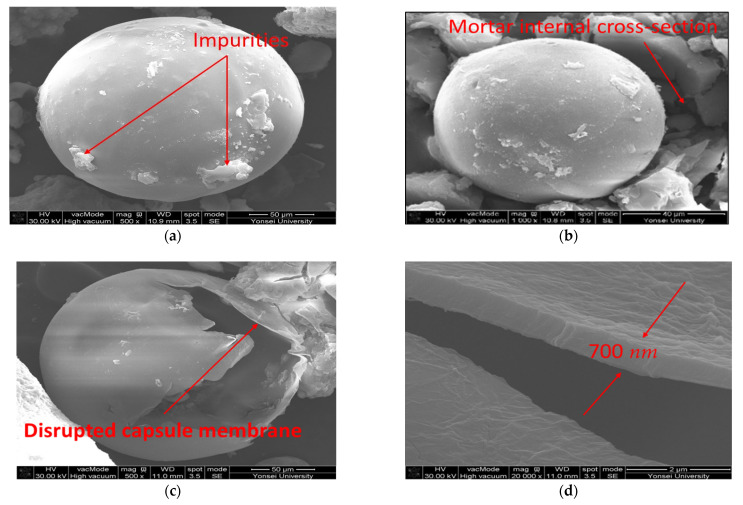
Shape of MCs (SEM): (**a**) original MCs (×500); (**b**) MCs inside mortar (×500); (**c**) broken MCs (×500); (**d**) membrane thickness of MCs (×20,000).

**Figure 6 materials-15-08907-f006:**
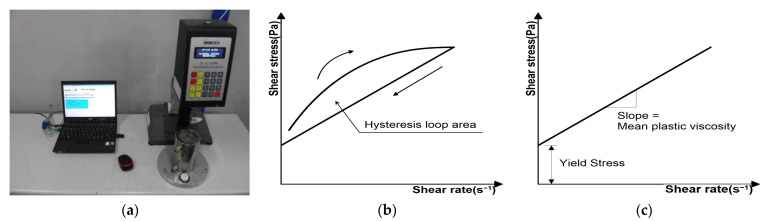
Rheological analysis method: (**a**) mortar viscometer; (**b**) hysteresis curve; (**c**) bingham model.

**Figure 7 materials-15-08907-f007:**
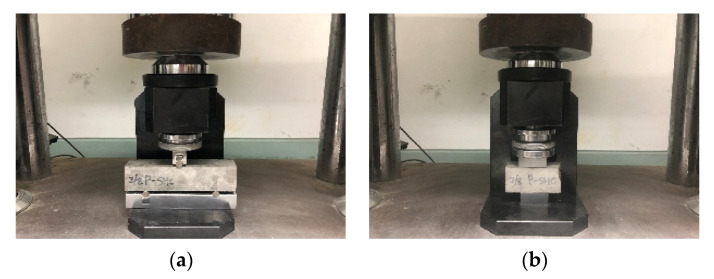
Experiment view: (**a**) flexural strength test; (**b**) compressive strength test.

**Figure 8 materials-15-08907-f008:**
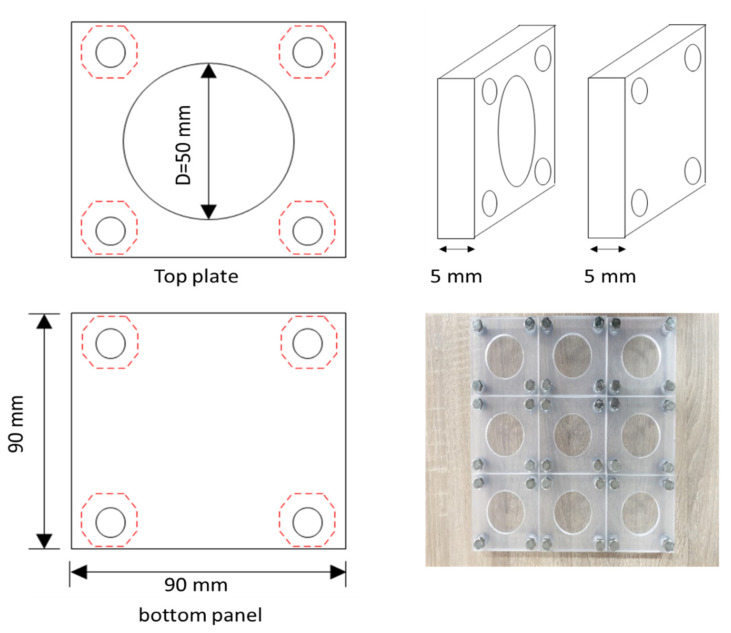
Water flow test specimens.

**Figure 9 materials-15-08907-f009:**
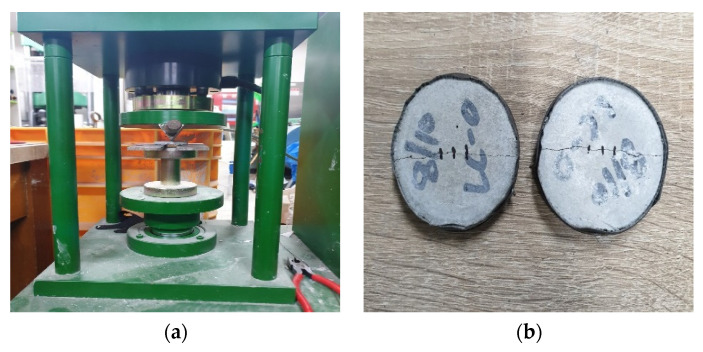
Crack induction and control process: (**a**) crack induction; (**b**) crack control.

**Figure 10 materials-15-08907-f010:**
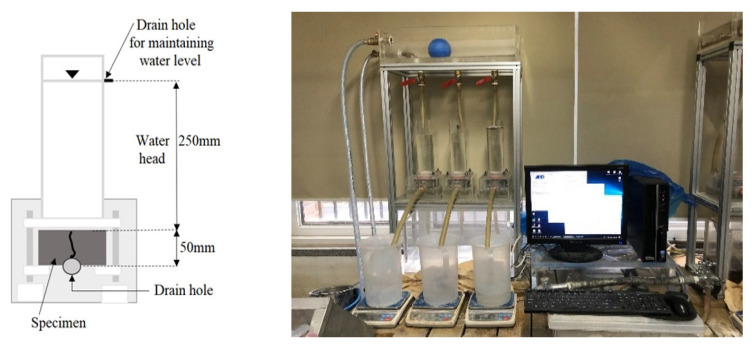
Water flow test.

**Figure 11 materials-15-08907-f011:**
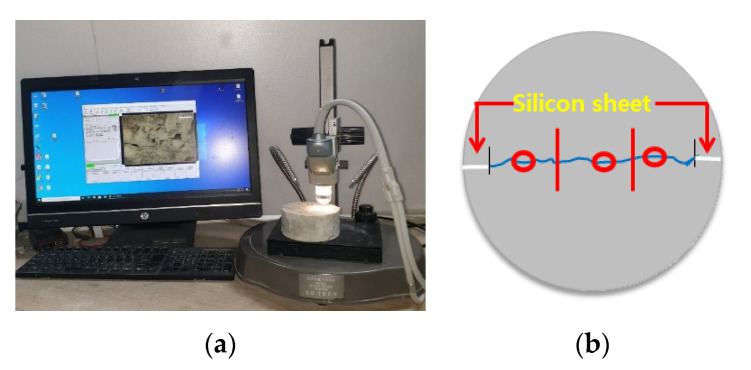
Crack closing test: (**a**) crack measuring equipment; (**b**) crack observation location.

**Figure 12 materials-15-08907-f012:**
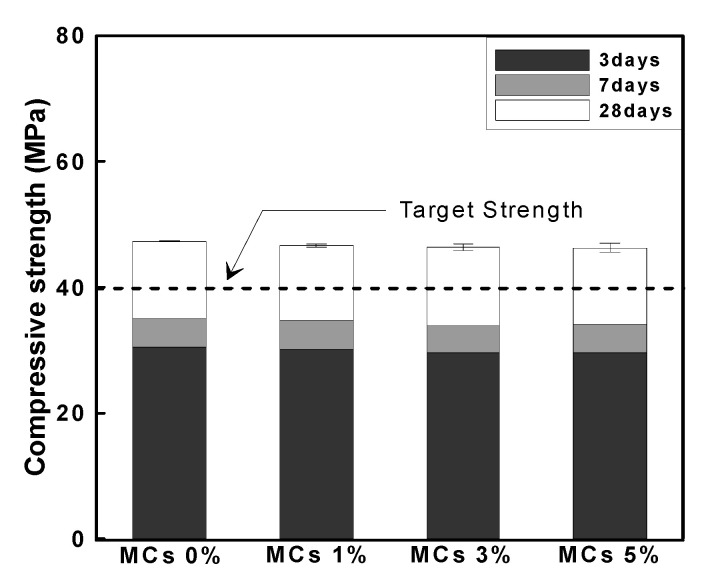
Evaluation result of compressive strength according to age.

**Figure 13 materials-15-08907-f013:**
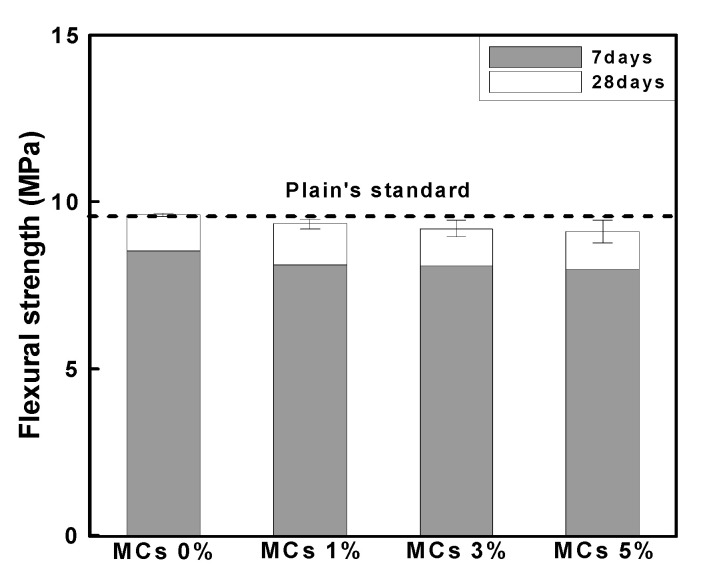
Evaluation result of flexural strength according to MCs.

**Figure 14 materials-15-08907-f014:**
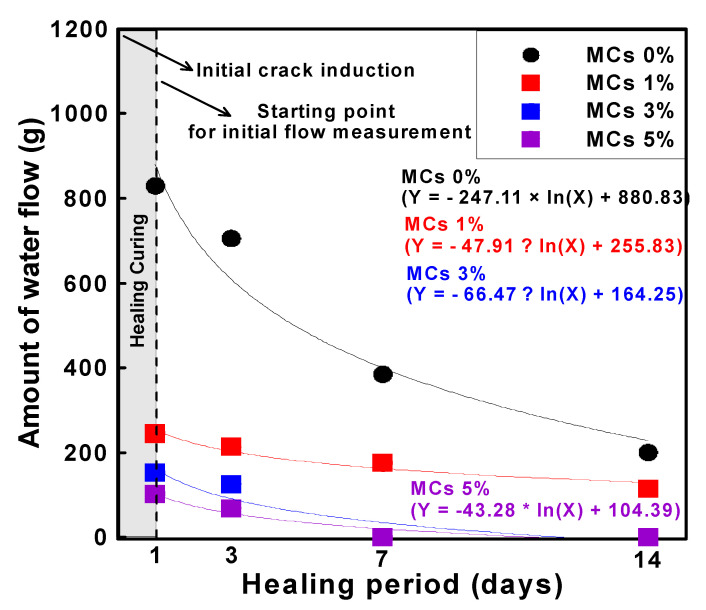
Variation healing rate according to MCs mixing rate (3 to 14 days for 1 day).

**Figure 15 materials-15-08907-f015:**
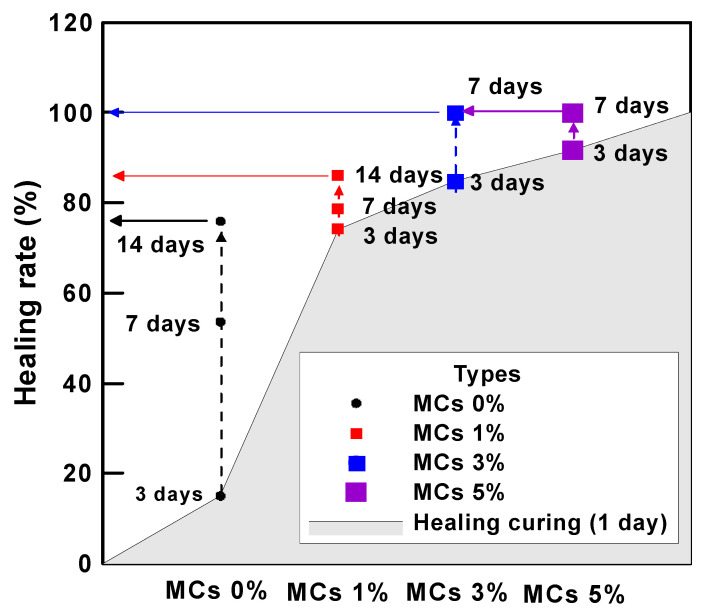
Time required for healing for each type.

**Figure 16 materials-15-08907-f016:**
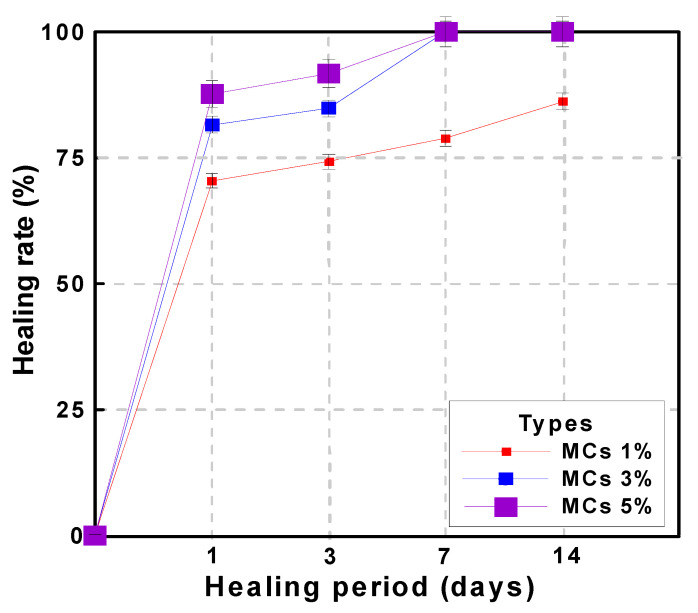
Correlation between healing period and healing rate on the MCs 0%.

**Figure 17 materials-15-08907-f017:**
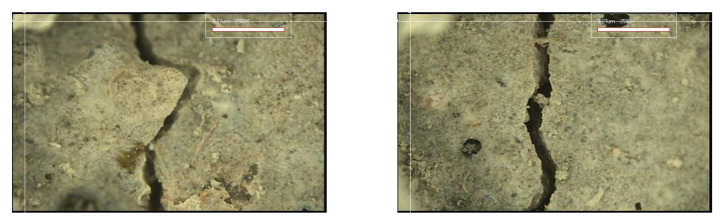
Surface crack monitoring of MCs 0%.

**Figure 18 materials-15-08907-f018:**
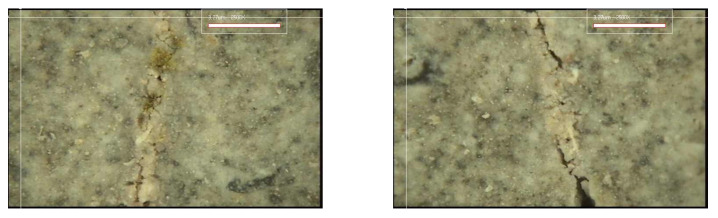
Surface crack monitoring of MCs 1%.

**Figure 19 materials-15-08907-f019:**
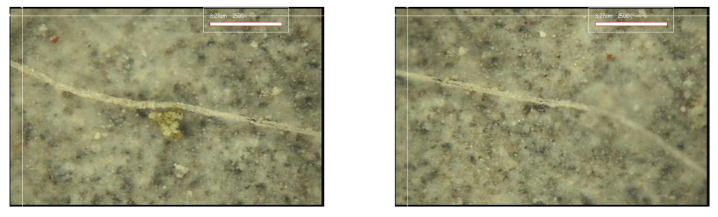
Surface crack monitoring of MCs 3%.

**Figure 20 materials-15-08907-f020:**
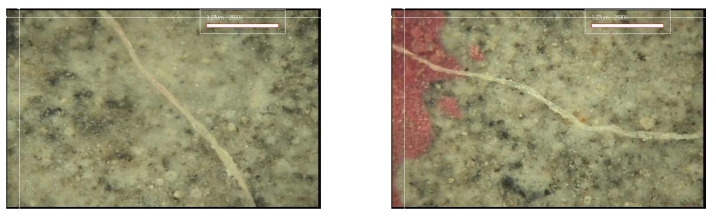
Surface crack monitoring of MCs 5%.

**Figure 21 materials-15-08907-f021:**
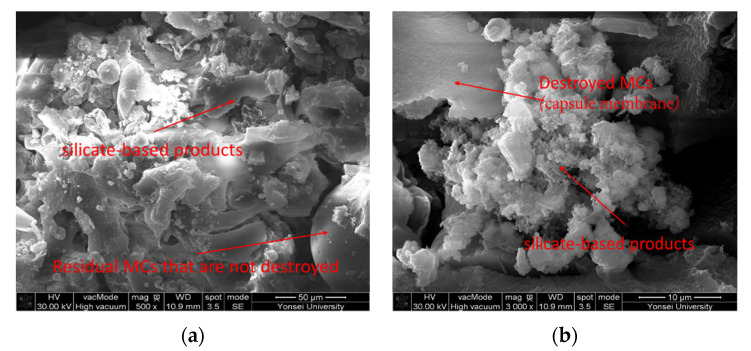
Observation result of healing area: (**a**) ×500; (**b**) ×3000.

**Table 1 materials-15-08907-t001:** Physical properties and chemical components of each silicate.

Item	Specific Gravity (20 °C)	K_2_O (%)	Na_2_O (%)	Li_2_O (%)	SiO_2_ (%)	Fe_2_O_3_ (%)	Mole Fraction	Viscosity (Cps, 20 °C)	Solids Content (%)
Potassium silicates	1.27–1.29	10.0–11.0	-	-	21.5–22.5	0.05	3.2–3.5	≤20	20–52
Sodium silicates	≥1.38	-	9.0–10.0	-	28.0–30.0	0.03	3.10–3.30	-	30–56
Lithium silicates	1.15–1.20	-	-	1.0–1.5	18.0–22.0	-	7.5–8.5	-	20–25

**Table 2 materials-15-08907-t002:** Mix ratio of mortar.

Type	W	C	S	MCs (C × %)	Ad. (C × %)
MCs 0%	0.4	1	2	0	0.5–0.6
MCs 1%	0.4	1	2	1	
MCs 3%	0.4	1	2	3	
MCs 5%	0.4	1	2	5	

**Table 3 materials-15-08907-t003:** Particle size distribution of ISO standard sand (ISO 679).

Size of Sieve (mm)	2	1.6	1.0	0.5	0.16	0.08
cumulative residue of sieve (%)	0	7 ± 5	33 ± 5	67 ± 5	87 ± 5	99 ± 1

**Table 4 materials-15-08907-t004:** Rheological parameters by the Bingham model.

Types	MCs				Relative Ratio (%)		
Rheological Parameters	0% (Plain)	1%	3%	5%	1%	3%	5%
Plastic viscosity (Pa·s)	24.5	22.2	20.6	18.1	−9.4	−15.9	−26.1
Yield stress (Pa)	211	152	165	174	−28.0	−21.8	−17.5

**Table 5 materials-15-08907-t005:** Test results of flow and air content according to MCs.

Types	MCs				Relative Ratio (%)		
	0% (Plain)	1%	3%	5%	1%	3%	5%
Flow (mm)	180	170	160	150	−5.6	−11.1	−16.7
Air content (%)	6.0	6.1	5.9	5.8	+1.7	−1.7	−3.3

**Table 6 materials-15-08907-t006:** Healing performance according to MCs.

Types	Water Flow (g) according to the Healing Period (Days)				Healing Rate (Equation (2))		
	1	3	7	14	3	7	14
MCs 0%	829	705	384.75	200.1	15.0	53.6	75.9
MCs 1%	246	214	176	115	74.2	78.8	86.1
MCs 3%	153	126.2	0	0	84.8	100.0	100.0
MCs 5%	102.6	69	0	0	91.7	100.0	100.0

## Data Availability

Not applicable.

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
