# Peer review of "A Study on the Healing Performance of Mortar with Microcapsules Using Silicate-Based Inorganic Materials"

_materials, 2022, doi:10.3390/ma15248907_

Round 1

Reviewer 1 Report

The paper focuses on the effects of self-healing microcapsules on the quality and healing properties of cement composites. A silicate-based inorganic material mixture was used as the healing material tested with ordinary Portland cement. And the effects of MCs on the rheological, mechanical, and healing properties of cement composites were determined. The authors have performed detailed experimental investigations about the claims they have made in the abstract. The paper falls within the circle of Materials based on the theme, contents and patterns of experimental investigations performed. I can recommend the paper for possible consideration/publication, but for a journal like Materials with high repute in basic as well as applied research in Materials industry, the manuscript needs to deal and explain many aspects and hence requires considerable improvements before publication in Materials. I have the following observations. If the authors agree to modify and improve the manuscript in light of my observations, I can recommend it for publications.

1. The whole article is difficult to understand, and the level of English writing needs to be improved and should be revised.

2. Literature review is not adequate and please reference more studies.

3. Please check the Instructions for Authors about the format of figure.

4. The case of some words is incorrect. Some professional words should be indicated by the full name when they first appear.

5. Please circle out the significant part in Figure 5.

6. Since the water flow decreased as the healing period increased when the MCs were mixed,how the authors taking such a mechanism into account for explaining figure 14 and 15.

7. The mechanical behaviors of microcapsule itself need further investigation to ensure an appropriate break time.

Author Response

November 23, 2022

Dear Reviewer,

Thank you very much for your valuable comments. I, with my co-authors, carefully revised the manuscript entitled “A Study on the Healing Performance of Mortar with Micro-capsules Using Silicate-based Inorganic Materials” according to the reviewer’s comments as follows.

Comment 1: The whole article is difficult to understand, and the level of English writing needs to be improved and should be revised.

Response 1: Extensive revisions were made throughout the manuscript, and English grammar and writing were reviewed.

Comment 2: Literature review is not adequate and please reference more studies.

Response 2: The introduction has been entirely revised and rewritten to a large extent, citing various literature.

Comment 3: Please check the Instructions for Authors about the format of figure.

Response 3 The journal template was checked, and the picture format was improved according to the standard.

Comment 4: The case of some words is incorrect. Some professional words should be indicated by the full name when they first appear.

Response 4: Circles have been added to important parts of the picture, and necessary explanations have been supplemented.(Line 164)

Comment 5: Please circle out the significant part in Figure 5.

Response 5: Some word errors in the manuscript were checked and revised. In addition, specialized words were indicated by full names when first introduced.

Comment 6: Since the water flow decreased as the healing period increased when the MCs were mixed,how the authors taking such a mechanism into account for explaining figure 14 and 15.

Response 6: The reaction mechanism was supplemented in the manuscript, and the explanation of the figure was supplemented.(changed Figure 15, Figure 16, New Figure 14)

Comment 7: The mechanical behaviors of microcapsule itself need further investigation to ensure an appropriate break time.

Response 7: The mechanical properties of the microcapsule itself were not addressed in this manuscript. In the future, we plan to conduct research considering the characteristics of the microcapsule itself, and plan to submit a new thesis by organizing the contents.

Thank you for reviewing the paper. We will finalize the manuscript by reflecting the opinions of all reviewers along with your comments.

Reviewer 2 Report

The manuscript deals with the healing performance of mortar with microcapsules using silicate-based inorganic materials. The topic of self-healing of concrete is very relevant nowadays - many workplaces deal with it and there are many articles on the subject. Therefore, one of the shortcomings of this manuscript is that the literature section cites only a small number of articles on this topic (self-healing of concrete) - I explain this in detail below. Otherwise, the article can be described as interesting and original. The methods are described in detail, including photo documentation - I appreciate that. I have the following comments on the manuscript:

 1) As mentioned above, the manuscript cites an insufficient number of articles on the subject. It is necessary to comment on them and compare them with the research described in the manuscript - especially in terms of the effectiveness of each method. I recommend adding at least 10 more articles on this topic (self-healing of concrete) to the literature, especially the following:

- Roig-Flores, M.; Moscato, S.; Serna, P.; Ferrara, L. Self-healing capability of concrete with crystalline admixtures in different environments. Constr. Build. Mater. 2015, 86, 1–11. DOI: 10.1016/j.conbuildmat.2015.03.091

- Zakova H., Pazderka J., Reiterman P.: Textile Reinforced Concrete in Combination with Improved Self-Healing Ability Caused by Crystalline Admixture. Materials. Vol. 13(24), a. no. 5787, 2020. DOI: 10.3390/ma13245787

- Sisomphon, K.; Copuroglu, O.; Koenders, E.A.B. Self-healing of surface cracks in mortars with expansive additive and crystalline additive. Cem. Concr. Compos. 2012, 34, 566–574. DOI: 10.1016/j.cemconcomp.2012.01.005

- other at least 7 more articles on the topic of concrete self-healing (there are many)

2) Paragraph no. 2 in the Conclusion contains the sentence:These results indicate that the influence of the moisture content in the microcapsules is more dominant than the influence of the mixing rate of the microcapsules.”. This is a very interesting knowledge, but it can fundamentally affect the effectiveness of the described method. It is necessary to elaborate this much more, this means to specify it more, to describe the mutual relationship, etc.

3) In the paragraph no. 3 in the Conclusion, there it is written that: “….the healing rate was more than 95% after 7 d of healing.”. It is not clear what the 95% refers to? Please explain it in detail to make the benefit of the technology clear.

Conclusion:

After incorporating the comments into the text (according to the above instructions) I recommend the manuscript for publication.

Author Response

November 23, 2022

Dear Reviewer,

Thank you very much for your valuable comments. I, with my co-authors, carefully revised the manuscript entitled “A Study on the Healing Performance of Mortar with Micro-capsules Using Silicate-based Inorganic Materials” according to the reviewer’s comments as follows.

Comment 1: As mentioned above, the manuscript cites an insufficient number of articles on the subject. It is necessary to comment on them and compare them with the research described in the manuscript - especially in terms of the effectiveness of each method. I recommend adding at least 10 more articles on this topic (self-healing of concrete) to the literature, especially the following:

   - Roig-Flores, M.; Moscato, S.; Serna, P.; Ferrara, L. Self-healing capability of concrete with crystalline admixtures in different environments. Constr. Build. Mater. 2015, 86, 1–11. DOI: 10.1016/j.conbuildmat.2015.03.091

   - Zakova H., Pazderka J., Reiterman P.: Textile Reinforced Concrete in Combination with Improved Self-Healing Ability Caused by Crystalline Admixture. Materials. Vol. 13(24), a. no. 5787, 2020. DOI: 10.3390/ma13245787

   - Sisomphon, K.; Copuroglu, O.; Koenders, E.A.B. Self-healing of surface cracks in mortars with expansive additive and crystalline additive. Cem. Concr. Compos. 2012, 34, 566–574. DOI: 10.1016/j.cemconcomp.2012.01.005

   - other at least 7 more articles on the topic of concrete self-healing (there are many)

Response 1: The manuscript has been extensively revised throughout. The introduction used a lot of literature, and the manuscript was revised including the literature you suggested.(References 17, 18, 19, Line 47) A total of 57 documents were cited by adding 40 from the existing 17 documents.

Comment 2: Paragraph no. 2 in the Conclusion contains the sentence: “These results indicate that the influence of the moisture content in the microcapsules is more dominant than the influence of the mixing rate of the microcapsules.”. This is a very interesting knowledge, but it can fundamentally affect the effectiveness of the described method. It is necessary to elaborate this much more, this means to specify it more, to describe the mutual relationship, etc.

Response 2: The “Effect of Moisture in Microcapsules” mentioned in Conclusion 2 of the original manuscript was deleted because there was no clear relevant evidence. In this part, additional research will be conducted in the future and a new manuscript presented quantitatively will be submitted.(Line 374)

Comment 3: In the paragraph no. 3 in the Conclusion, there it is written that: “….the healing rate was more than 95% after 7 d of healing.”. It is not clear what the 95% refers to? Please explain it in detail to make the benefit of the technology clear.

Response 3: Conclusion 3 of the original manuscript was newly summarized and organized after a sentence error was found.(Line 380)

Thank you for reviewing the paper. We will finalize the manuscript by reflecting the opinions of all reviewers along with your comments.

Reviewer 3 Report

Overall, whilst there are some interesting results presented in the manuscript, it does not meet the standards of the journal and has some serious flaws. My main concerns and comments are given below:

·         The introduction is too short and does not give a good overview of the field, nor identify the research gap. From the introduction it seems that the research gap is the use of inorganic healing agents, however, a number of researchers have used inorganic healing agents, including silicates (sodium silicate is commonly used) that are also the focus of this work.

·         It is unclear which healing agent was used in the study; in Equation 1 the healing agent is given as A2SiO2, what is A?

·         A number of healing agents target self-sealing vs self-healing and a regain in mechanical properties is not observed. However, when silicate based healing agents, such as sodium silicates, are used, a good degree of mechanical healing can be achieved (~=30%). As the authors are using a silicate based healing agent, why did they not consider the mechanical healing in their investigation?

·         What was the mixing rate for the microcapsules based on (i.e. is it % by weight or by volume)?

·         A number of figures in the manuscript are either unhelpful, poor quality or both. For example, Figure 1 (a), Figures 6, 7 (on page 10), 14 & 15 are poor quality; whilst Figures 7 (on page 5) and 5 (a) are unhelpful and do not add to the manuscript. In addition, the Figure numbers are repeated at times, and not in order at others. For example, the Figures are numbered (in the order they appear in the paper) as 1-10, 5-7, 14-15, 8, 17-19.

·         Limited information on the microcapsules has been provided, for example, what is the average size and shell thickness (and their variability)? In addition, details of the production are limited, whilst the steps are given in Figure 3, there are no details given, for example what are the solutions used? How were the capsules polymerised? What is the reinforcement coating?

·         The authors state that increasing the mixing ratio of microcapsules has no significant effect on the yield stress of the fresh mortar sample, however, Table 2 shows a 10% difference between the 1 and 5% cases. Presumably this difference would increase with increasing mixing ratio and is not insignificant?

·         When discussing the mechanical results, the effect of water was discussed as the reason for the differences. Do the authors mean that the amount of water added to the mix was reduced as the microcapsules themselves were used in a wet state (thereby providing some water)? Do the authors expect that at higher mixing rates the mechanical properties will be reduced? Usually, there is a decrease in mechanical properties associated with the introduction of microcapsules, do the authors suggest that this decrease seen in the literature is due to the lack of consideration of the effect of water?

·         The water flow tests show a high amount of healing after just one day when compared to the reference sample, what do the authors attribute this too? Is it due to the healing reaction, or could the capsules themselves be blocking the crack (perhaps having swelled through water absorption)?

Author Response

November 23, 2022

Dear Reviewer,

Thank you very much for your valuable comments. I, with my co-authors, carefully revised the manuscript entitled “A Study on the Healing Performance of Mortar with Microcapsules Using Silicate-based Inorganic Materials” according to the reviewer’s comments as follows.

Comment 1: The introduction is too short and does not give a good overview of the field, nor identify the research gap. From the introduction it seems that the research gap is the use of inorganic healing agents, however, a number of researchers have used inorganic healing agents, including silicates (sodium silicate is commonly used) that are also the focus of this work.

Response 1: The manuscript has been extensively revised in the first half. A number of references have been cited and rewritten.

Comment 2: It is unclear which healing agent was used in the study; in Equation 1 the healing agent is given as A2SiO2, what is A?

Response 2: 'A' is a symbol that collectively refers to strong alkali ions. The manuscript did not explain this, so it was supplemented.(Line 143)

Comment 3: A number of healing agents target self-sealing vs self-healing and a regain in mechanical properties is not observed. However, when silicate based healing agents, such as sodium silicates, are used, a good degree of mechanical healing can be achieved (~=30%). As the authors are using a silicate based healing agent, why did they not consider the mechanical healing in their investigation?

Response 3: Mechanical effects were not addressed in this manuscript. The reason is that there are too many data, so it was difficult to include them in one manuscript. In the future, we plan to submit a new manuscript about the mechanical effect.

Comment 4: What was the mixing rate for the microcapsules based on (i.e. is it % by weight or by volume)?

Response 4: The mixing ratio of microcapsules is based on the cement mass. The authors confirmed what was not explained in the manuscript and supplemented the contents.(Line 172)

Comment 5:   A number of figures in the manuscript are either unhelpful, poor quality or both. For example, Figure 1 (a), Figures 6, 7 (on page 10), 14 & 15 are poor quality; whilst Figures 7 (on page 5) and 5 (a) are unhelpful and do not add to the manuscript. In addition, the Figure numbers are repeated at times, and not in order at others. For example, the Figures are numbered (in the order they appear in the paper) as 1-10, 5-7, 14-15, 8, 17-19.

Response 5: The error in the caption number was checked in the manuscript and corrected correctly. In addition, pictures of poor quality were replaced.

Comment 6: Limited information on the microcapsules has been provided, for example, what is the average size and shell thickness (and their variability)? In addition, details of the production are limited, whilst the steps are given in Figure 3, there are no details given, for example what are the solutions used? How were the capsules polymerised? What is the reinforcement coating?

Response 6: The reason for the lack of information on microcapsules was to substitute references due to the space of the manuscript. However, the authors also agreed that the information was very lacking, and supplemented the information on microcapsules.(Line113-165)

Comment 7: The authors state that increasing the mixing ratio of microcapsules has no significant effect on the yield stress of the fresh mortar sample, however, Table 2 shows a 10% difference between the 1 and 5% cases. Presumably this difference would increase with increasing mixing ratio and is not insignificant?

Response 7: Errors in the manuscript were identified and revised.(Line 248-261)

Comment 8: When discussing the mechanical results, the effect of water was discussed as the reason for the differences. Do the authors mean that the amount of water added to the mix was reduced as the microcapsules themselves were used in a wet state (thereby providing some water)? Do the authors expect that at higher mixing rates the mechanical properties will be reduced? Usually, there is a decrease in mechanical properties associated with the introduction of microcapsules, do the authors suggest that this decrease seen in the literature is due to the lack of consideration of the effect of water?

Response 8: The “Effect of Moisture in Microcapsules” mentioned of the original manuscript was deleted because there was no clear relevant evidence. In this part, additional research will be conducted in the future and a new manuscript presented quantitatively will be submitted.(Line 374)

Comment 9: The water flow tests show a high amount of healing after just one day when compared to the reference sample, what do the authors attribute this too? Is it due to the healing reaction, or could the capsules themselves be blocking the crack (perhaps having swelled through water absorption)?

Response 9: The experimental results of the water flow test were revised in more detail. Reaction mechanisms and mechanisms have been mentioned.(Line 299-322)

In addition, SEM pictures of the healing product were added to confirm that the filling in the crack area was not a capsule. This has also been added to the text. .(Figure 21)

Thank you for reviewing the paper. We will finalize the manuscript by reflecting the opinions of all reviewers along with your comments.

Round 2

Reviewer 3 Report

Overall, my main concerns about the manuscript have been addressed. The introduction has been greatly expanded. The novelty of the work is now clearer (which includes for example the use of a healing agent comprised Potassium, Sodium and Lithium silicates). In addition, a number of things that were unclear/not been presented in the original manuscript (such as the microcapsule properties), have been clarified or expanded on. In addition, the presentation and discussion of the results has been improved.

The manuscript in its current form is greatly improved and now meets the standards for the journal. Therefore, I recommend that it is accepted in it's current form.